# Overexpression of Transcripts Coding for Renin-b but Not for Renin-a Reduce Oxidative Stress and Increase Cardiomyoblast Survival under Starvation Conditions

**DOI:** 10.3390/cells10051204

**Published:** 2021-05-14

**Authors:** Heike Wanka, Philipp Lutze, Alexander Albers, Janine Golchert, Doreen Staar, Jörg Peters

**Affiliations:** 1Institute of Physiology, University Medicine Greifswald, 17475 Greifswald, Germany; Heike.Wanka@med.uni-greifswald.de (H.W.); lutze@fbn-dummerstorf.de (P.L.); Alex.Albers@gmx.de (A.A.); Janine.Golchert@med.uni-greifswald.de (J.G.); doreen.staar@med.uni-greifswald.de (D.S.); 2Leibniz Institute for Farm Animal Biology (FBN), Institute of Muscle Biology & Growth, Wilhelm-Stahl-Allee 2, 18196 Dummerstorf, Germany

**Keywords:** cardiac H9c2 cells, cytosolic renin, secretory renin, ischemia-induced cell death, reactive oxygen species, mitochondrial membrane potential, ATP levels

## Abstract

A stimulated renin-angiotensin system is known to promote oxidative stress, apoptosis, necrosis and fibrosis. Renin transcripts (renin-b; renin-c) encoding a cytosolic renin isoform have been discovered that may in contrast to the commonly known secretory renin (renin-a) exert protective effects Here, we analyzed the effect of renin-a and renin-b overexpression in H9c2 cardiomyoblasts on apoptosis and necrosis as well as on potential mechanisms involved in cell death processes. To mimic ischemic conditions, cells were exposed to glucose starvation, anoxia or combined oxygen–glucose deprivation (OGD) for 24 h. Under OGD, control cells exhibited markedly increased necrotic and apoptotic cell death accompanied by enhanced ROS accumulation, loss of mitochondrial membrane potential and decreased ATP levels. The effects of OGD on necrosis were exaggerated in renin-a cells, but markedly diminished in renin-b cells. However, with respect to apoptosis, the effects of OGD were almost completely abolished in renin-b cells but interestingly also moderately diminished in renin-a cells. Under glucose depletion we found opposing responses between renin-a and renin-b cells; while the rate of necrosis and apoptosis was aggravated in renin-a cells, it was attenuated in renin-b cells. Based on our results, strategies targeting the regulation of cytosolic renin-b as well as the identification of pathways involved in the protective effects of renin-b may be helpful to improve the treatment of ischemia-relevant diseases.

## 1. Introduction

Alternative renin transcripts, lacking exon 1, have been identified in several tissues, including the heart of rats (termed exon1a renin) [1], in the brain of mice (termed renin-b) [2] as well as in the brain (renin-b) or lungs (renin-c) of transgenic mice expressing a human renin gene construct [3]. In the rat heart, this transcript, termed renin-b, is under the control of an alternative promoter located in intron 1 [4]. In cardiac cells, the promoter is stimulated by glucose depletion in a serum response factor-dependent manner [4]. Furthermore, transcript levels of renin-b, but not of the commonly known secretory renin (renin-a), increased markedly after myocardial infarction in vivo [5]. Due to the absence of the signal for a co-translational transport to the endoplasmic reticulum encoded by exon 1, the alternative renin transcripts are translated at free ribosomes [1,2,3]. The translated protein represents a truncated prorenin, which is found in the cytosol as well as within mitochondria [1,6,7,8].

We previously overexpressed the coding region of renin-b without its 5′ untranslated region, which is derived from intron A of the renin gene in H9c2 cells. These cells were protected from necrotic cell death under glucose starvation [9] and from necrotic as well as apoptotic cell death induced by oxygen combined with glucose depletion (OGD) [10]. They also exhibited a switch to a more aerobic glycolysis, known as the Warburg effect, which may be favorable under ischemic conditions [11]. Renin-b also reduced levels of reactive oxygen species (ROS) [10]. Excessive ROS production leads to cell death in cardiac and other tissues. Elevated mitochondrial ROS disrupt the mitochondrial membrane potential, thereby leading to mitochondrial swelling, insufficiency to produce ATP, opening of the mitochondrial permeability transition pore, release of ROS and of pro-apoptotic factors into the cytosol and finally apoptotic and necrotic cell death [12,13,14,15]. However, neither the role of the full-length transcript encoding renin-b, nor the role of renin-a, when overexpressed in cardiac cells, are known. To support the hypothesis of an existing endogenously protective renin-b, it was now essential to demonstrate that exactly this transcript, which is found in vivo, exerts the same protective effects as overexpression of exon (2–9), lacking exon1a and hence its 5′UTR. Therefore, our aims were (1) to investigate the effects of renin-a overexpression in H9c2 cells in general, and (2) to demonstrate that the already observed protective effects of artificial exon (2–9) renin overexpression were still present when the endogenous renin-b was used, including its 5′UTR upstream of exons 2-on necrosis, apoptosis, and the production of reactive oxygen species under starvation conditions such as glucose depletion and anoxia.

## 2. Materials and Methods

### 2.1. Cell Culture

The H9c2 cell line obtained from American Type Culture Collection (ATCC, Manassas, VA, USA) was maintained in DMEM medium supplemented with 100 U/mL penicillin, 100 µg/mL streptomycin and 10% fetal bovine serum under a humidified atmosphere of 5% CO_2_ at 37 °C. Media exchange was performed every 3 days and cells were sub-cultured after having reached around 80% confluence. Glucose concentration was 25 mM according to ATCC suggestion.

H9c2 cells were transfected as previously described [8] with the empty pIRES vector or with the pIRES vector containing the coding region of renin-a or the coding region of renin-b together with its 5′UTR. Upregulation of renin-b mRNA and renin-a mRNA was 5-fold and 10-fold, respectively, as determined by qRT-PCR analysis. Thus, renin-b mRNA normalized to Ywhz was in pIRES control cells 2 × 10^−5^ and in renin-b cells 1 × 10^−4.^ and renin-a mRNA normalized to Ywhz was in pIRES control cells 4 × 10^−5^ and in renin-a cells: 4 × 10^−4^.

To ensure overexpression of renin, the transfected pIRES control cells (empty vector) as well as the renin-b and renin-a overexpressing cell lines were cultured in the presence of 430 µg/mL G418 sulfate. For functional analyses, pIRES control, renin-a and renin-b cells were seeded in 6-well or 96-well culture plates, respectively, for 72 h. They were then exposed to control conditions, glucose starvation, anoxia (AnaeroPack rectangular jar, Mitsubishi Gas Chemical Company Inc, Tokyo, Japan; GENbox anaer, Biomerieux, Craponne, France) or the combination of oxygen and glucose depletion, for 24 h at 5% CO_2_ and 37 °C, followed by RT-qPCR analyses to detect renin transcript abundances, detection of cell death and analyses of mitochondrial parameters.

### 2.2. Quantitative PCR

RNA was extracted as described previously [10]. The following sets of primers were used to determine renin-a, renin-b and Ywhz mRNA respectively (Table 1). The cycle threshold (Ct) number in combination with the 2−ΔΔCT method was used to detect changes in renin transcript expression compared to Ywhaz.

### 2.3. Necrotic Cell Death

For determination of necrosis, 1 × 10^4^ cells in 100 µL medium were seeded in 96-well plates as a six-fold attempt. Three wells were used for the detection of spontaneous lactate dehydrogenase (LDH) release and the other 3 wells for the determination of the total cellular LDH content. After a 72 h growing phase at 37 °C and 5% CO_2_ followed by incubation under ischemia-related conditions for another 24 h, necrosis rate was analyzed using the Cytotoxicity Detection Kit (LDH) (Sigma-Aldrich, St Louis, MO, USA) as previously described [8]. The necrosis rate was calculated by normalizing the amount of released LDH to the LDH content of the cells.

### 2.4. Apoptosis, ROS- and ATP-Levels, Mitochondrial Membran Potential

Apoptosis, ROS levels and inner mitochondrial membrane potential and ATP levels were determined as described previously [10,16].

### 2.5. Statistical Analyses

The data presented are group means ± SEM of independently performed experiments. To specify differences between groups, two-way ANOVA with Bonferroni correction were performed as appropriate using GraphPad Prism (Graph Pad Software, La Jolla, CA, USA). *p*-values below 0.05 were considered statistically significant.

## 3. Results

### 3.1. Renin-b Overexpression Attenuates Loss of Viability under OGD

H9c2 cells were exposed to control or various ischemia-related conditions. We found that, under basal conditions, the growth rate of renin-b cells was reduced compared to pIRES control cells (*p* < 0.05) (Figure 1A). However, the membrane integrity of renin-b cells was unchanged as detected initially by the trypan blue exclusion test (Figure 1B). Exposure of cells to glucose starvation had no effect on these parameters, while anoxia alone induced a reduction of pIRES cell number (*p* < 0.01) and a decrease of membrane integrity in all cell lines (pIRES: *p* < 0.001; renin-b: *p* < 0.05 and renin-a cells: *p* < 0.05). In contrast, OGD caused a marked decrease in cell numbers in pIRES (*p* < 0.001) and renin-a cells (*p* < 0.001), but not in renin-b cells. In comparison, the membrane integrity was reduced in all three cell lines.

These data suggest that renin-b cells may be protective during exposure to ischemia-related conditions. The marked OGD-induced decrease in cell number and the loss of membrane integrity indicates a vigorous cell damage probably due to a necrotic cell death in pIRES and renin-a cells from which renin-b cells seems to be protected.

### 3.2. Renin-b but Not Renin-a Overexpression Protects from Ischemia-Induced Necrosis and Apoptosis

Next, we ascertained to which magnitude necrosis and apoptosis were involved in the OGD-induced cell death. For the early stage of necrosis, we analyzed the loss of membrane integrity of the cells, when cell morphology was still recognizable (Figure 2A). For the late stage of necrosis, we analyzed the release of the lactate dehydrogenase (LDH) (Figure 2B), when cells were already collapsed.

Under basal conditions, the number of propidium iodide (PI)-positive, early necrotic cells, was higher in renin-b cells (*p* < 0.05) compared to pIRES control and renin-a cells. The amount of late necrosis as detected by the LDH ratio was low and similar between the cell lines. Exposure to glucose starvation affected only the degree of late necrosis. The LDH ratio increased significantly in pIRES control (*p* < 0.01) and even more pronounced in renin-a cells (*p* < 0.001), while renin-b cells remained unaffected. Anoxia alone was well tolerated by the cell lines. Only in pIRES control cells, the percentage of early, PI-positive necrotic cells increased significantly (*p* < 0.01). OGD induced a marked increase of both early and late necrotic cell deaths in pIRES control (*p* < 0.001) and renin-a cells (*p* < 0.001). Late necrosis was also increased in renin-b cells (*p* < 0.01), but to a much lesser extent than in the other cell lines.

To quantify apoptotic cell death, we determined the involvement of the intrinsic and extrinsic pathways of apoptosis by analyzing the activation of caspases (intrinsic and extrinsic pathways), the binding of Annexin V (intrinsic pathway), and the amount of the Fas receptor (FasR, extrinsic pathway). By using PI together with one of the apoptosis markers, we separated early (PI-negative apoptosis-positive) from late apoptotic stages (PI-positive apoptosis-positive).

Under basal conditions, the total percentages of CaspACE-positive cells were similar between pIRES control, renin-b and renin-a cells (Figure 3A–C). Glucose starvation increased the percentages of total CaspACE-positive cells in pIRES controls (*p* < 0.01) and even more pronounced in renin-a cells (*p* < 0.01), but not in renin-b cells. These increases are due to the increase of early apoptotic cells, since the percentage of CaspACE-positive, PI-positive cells remained unchanged (Figure 3C).

Anoxia alone increased the percentage of total CaspACE-positive cells exclusively in pIRES control cells (*p* < 0.05). In contrast, in renin-b cells, the percentage of CaspACE-positive cells decreased significantly (*p* < 0.05) under anoxia. Exposure of pIRES control cells to OGD resulted in a marked increase of CaspACE-positive cells (*p* < 0.001), the increase being especially prominent in the late apoptotic stage (PI-positive CaspACE-positive) (*p* < 0.001). This OGD-induced increases in total as well as PI-positive CaspACE-positive pIRES cells (each *p* < 0.001) were partly reduced in renin-a cells (*p* < 0.001) and completely abolished in renin-b cells (*p* < 0.001).

Data of Annexin V binding as an indicator of the intrinsic, mitochondrial pathway of apoptosis are represented in Figure 4A–C. Under basal conditions, the percentage of total Annexin V-positive cells was increased in renin-b cells (*p* < 0.01) and in renin-a cells (*p* < 0.05) compared to pIRES control. Glucose starvation caused an increase in the percentage of total Annexin V-positive cells in pIRES control (*p* < 0.05) but not in renin-b cells. Again, this increase was due to an enhanced early apoptosis. Anoxia alone had no effect on the intrinsically triggered pathway of apoptosis in pIRES control and renin-a cells. In renin-b cells, the percentage of total Annexin V-positive cells decreased significantly (*p* < 0.05) under anoxia. Exposure to OGD caused a marked increase in the percentages of both total Annexin V-positive and PI-positive, Annexin V-positive cells in pIRES control (*p* < 0.001) and renin-a cells (*p* < 0.001). In contrast, in renin-b cells, the percentage of Annexin V-positiv cells remained at basal level, indicating a renin-b mediated protection from OGD-induced intrinsically triggered apoptosis.

Lastly, we analyzed the amount of FasR as an activating feature of the extrinsic pathway of apoptosis (Figure 5A–C). Under basal conditions, the amount of FasR was similar in all cell lines investigated. Exposure to glucose starvation increased the percentage of total FasR-positive cells in renin-a cells (*p* < 0.001), but not in pIRES control and renin-b cells. Anoxia alone caused an increase in the percentage of total and PI-positive FasR-positive cells exclusively in pIRES control cells (*p* < 0.05 and *p* < 0.001). Exposure to OGD resulted in a marked increase of extrinsically induced apoptosis. The percentages of total and PI-positive FasR-positive cells increased significantly in pIRES control (*p* < 0.001) and renin-a cells (*p* < 0.001), while in renin-b cells the amount of FasR remained unaffected.

Taken together, overexpression of renin-b but not of renin-a in H9c2 cells mediates high resistance to ischemia-induced damage, notably to OGD-induced necrotic and apoptotic cell death, whereby both the intrinsic and extrinsic pathway of apoptosis were effectively blocked.

### 3.3. Renin-b Overexpression Mitigates OGD-Induced Accumulation of ROS

Next, we asked whether renin-b could be protective by controlling the generation and elimination of reactive oxygen species (ROS) under ischemia-related conditions. By using the fluorophores MitoSOX and dihydroethidium (DHE) to detect mitochondrial superoxides and cytosolically localized ROS, respectively, we were able to identify the origin of dysregulated ROS management.

As seen in Figure 6, both renin overexpressing cell lines were characterized by a basal increase in levels of mitochondrial superoxides compared to pIRES control cells as indicated by the increase of the mean fluorescence intensity (FLI) of MitoSOX-positive cells (*p* < 0.05) (Figure 6A, B). The basal percentages of MitoSOX-positive cells were at the same level as in pIRES control cells (Figure 6C). Glucose starvation and anoxia alone had no effect on the management of mitochondrial superoxides. In comparison, OGD caused both an increase in the mean FLI and the percentages of MitoSOX-positive cells in pIRES control (*p* < 0.001) and renin-a cells (*p* < 0.001), respectively, indicating an accumulation of mitochondrial superoxides in these cell lines. Renin-b cells were protected from the OGD-induced increase in MitoSOX FLI. Furthermore, the increase in the percentage of MitoSOX-positive renin-b cells was mitigated (*p* < 0.001) compared to pIRES controls.

DHE labeling revealed that ROS are present in low concentrations in all analyzed cells (Figure 7A). The mean FLI of DHE-positive cells was at the same level in all investigated cell lines and remained stable under glucose starvation and anoxia alone compared to basal conditions (Figure 7A,B). Yet, a low percentage of cells exhibited an increased FLI of DHE in all cell lines during basal, glucose starvation, and anoxic conditions (Figure 7C). These cells, named DHEhigh-positive cells, reflect a dysregulated accumulation of ROS. The percentage of these dysregulated cells increased markedly after exposure to OGD in pIRES control cells (*p* < 0.001). In renin-a (*p* < 0.001), and more prominently in renin-b cells (*p* < 0.001), this increase was mitigated compared to pIRES control cells.

In summary, the mitochondrial and cytosolic ROS management is essential for avoiding ROS accumulation under anoxic conditions, where ROS generation should normally be increased in all investigated cell lines. During OGD, ROS management is no longer sufficient leading to ROS accumulation and cell death in pIRES control and renin-a cells. In contrast, the renin-b overexpression may cause the maintenance of mitochondrial and cytosolic ROS management and could be responsible for the increased cell survival during OGD in these cells.

### 3.4. Renin-b Overexpression Mitigates OGD-Induced Disruption of the Mitochondrial Membrane Potential and ATP Depletion

Another important factor for cell survival is the maintenance of the mitochondrial membrane potential (ΔΨm) associated with a sufficient production of ATP during ischemia-related conditions.

Under basal and glucose starvation conditions, the ΔΨm detected by the ratio of red to green fluorescence intensity of JC-1-positive cells (disturbed to intact ΔΨm) was at the same level in all cell lines investigated (Figure 8A). In cells exposed to anoxia alone, the ΔΨm increased significantly in pIRES control (*p* < 0.01) and more pronounced in renin-a cells (*p* < 0.001), while the ΔΨm of renin-b cells remained unaffected. In contrast, OGD caused a marked decrease in ΔΨm in pIRES control (*p* < 0.01) and renin-a cells (*p* < 0.01), but not in renin-b cells, indicating that renin-b overexpression may protect the cells from OGD-induced collapse of ΔΨm.

To substantiate the previous results, we finally quantified ATP levels of all cell lines (Figure 8B). Under basal conditions, renin-a cells exhibited reduced ATP levels (*p* < 0.001) compared to pIRES control and renin-b cells, respectively. Glucose starvation alone resulted in a significant decrease in ATP levels only in pIRES control cells (*p* < 0.05), while anoxia alone was generally associated with a marked decrease in ATP levels in all cell lines (*p* < 0.001 each). In comparison, the OGD-induced decrease in ATP levels was remarkable in all cell lines, but especially in pIRES controls (*p* < 0.001). In renin-a and renin-b cells, ATP levels decreased to a much lesser extent than in pIRES control cells (*p* < 0.001 each). Thereby, renin-b overexpression prevented the decrease in ATP levels under OGD more prominent than overexpression of renin-a.

Based on our results, the overexpression of cytosolic renin-b previous to an ischemic condition may be protective in the sense of an adaptation of the cells to the marked ATP deficiency during OGD.

## 4. Discussion

An alternative renin transcript, termed renin-b, recently aroused interest as a promising target to protect cardiac cells under ischemia-related conditions. In the rat heart, expression of renin-b but not of renin-a is increased after myocardial infarction [9]. In cardiac H9c2 cells, anoxia and OGD increased the expression of renin-b, but not of renin-a [10]. Furthermore, glucose starvation selectively increased the expression of renin-b in a serum response factor dependent manner [4,9]. These data indicate that upregulation of renin-b transcripts is involved in the response to ischemia-related factors. We previously demonstrated protective effects of the overexpressed coding region of renin-b under glucose depletion and OGD. Renin-b attenuated and even prevented the increase in necrosis and apoptosis rates as well as ROS generation. However, overexpressing only the coding region of renin-b does not reflect the true endogenous situation, since the endogenous renin-b mRNA is comprising its 5′ untranslated region (5′UTR; about 80 bases of intron 1). This 5′UTR likely has regulatory functions and may even prevent the translation of a renin-b protein. To support the hypothesis of an existing endogenous system based on renin-b we had to show that the full-length renin-b transcript including its 5′UTR exerts the same protective effects. Here, we chose a degree of overexpression (5-fold), which is also found endogenously under stimulatory conditions, such as glucose depletion, anoxia, or OGD [4,10,11]. The protective effects of renin-b were still seen in the presence of its 5′UTR. Thus, the protective effects were independently of the 5′UTR. When indirectly comparing the effects of renin-b overexpression and exon (2–9) renin overexpression from our previous study, the effects were almost identical with respect to necrosis, apoptosis, ATP, ΔΨm, and ROS levels. These data support our hypothesis of an existing endogenously protective system induced specifically by renin-b, but not by renin-a. We here need to consider, however, that the overexpression of renin-a was about twofold higher than the overexpression of renin-b. Thus, the effects are not directly comparable. Although we were not able to titrate down the overexpression of renin-a to exactly the level of overexpression of renin-b, we think it is unlikely that renin-a at lower dosage would be able to exert the same protective effects than renin-b but cannot exclude it completely.

In contrast to renin-a, the renin-b protein cannot be sorted to a secretory pathway. Instead, renin-b remains intracellular [2,7,8] and is imported into mitochondria [1]. The presence of renin within mitochondria has been shown by electron microscopy [6] and immunofluorescence [8] as well as enzymatically after separation of specific organelle fractions by differential centrifugation [6,7,8]. Therefore, it was intriguing to speculate, that renin-b modulates mitochondrial functions, either via intracellular angiotensin generation or via unknown mechanisms.

Mitochondria play a central role in cellular survival during ischemia and control the cellular energy metabolism [17,18]. Maintaining mitochondrial function is therefore an effective strategy to attenuate ischemia-induced injury of cardiac cells. Preserving ATP levels, attenuation of oxidative stress and mitochondrial Ca^2+^ overload as well as delayed correction of ischemia-induced intracellular acidosis are all events integrated in the ischemic preconditioning (IPC)-mediated inhibition of mitochondrial permeability pore (mPTP) opening [19,20]. These protective events resulting in reduced apoptosis and necrosis have been detected in H9c2 cells overexpressing the coding region of renin-b under OGD. The present study now demonstrates that the alternative renin-b transcript, when overexpressed at its full length, equally protects H9c2 cells from OGD-induced apoptosis, necrosis, and from loss of ΔΨm by mitigating the increase in ROS and decrease in ATP levels.

The harmful effects of renin-a are most likely explained by the production of angiotensin. H9c2 cells have the capability to produce angiotensin and respond to ANG II since they express the transcripts for all components of the renin–angiotensin system (RAS), such as angiotensinogen, angiotensin converting enzyme (ACE) and angiotensin receptors (AT1R; AT2R) [16]. Although the amount of ANG II may be rather small, ANG II may still reach considerably high concentrations locally. ANG II induces the expression of the Nox2 subunit of the NADPH-oxidase via AT1R, thereby increasing ROS production resulting in cardiac hypertrophy, inflammation and fibrosis [21]. Thus, our data agree with the hypothesis of an existing local cardiac RAS in H9c2 cells with the effector peptide ANG II and renin-a as the angiotensin generating enzyme. This local cardiac RAS has harmful cellular consequences, as demonstrated in the present study, i.e., increased necrosis and apoptosis under glucose depletion as well as increased necrosis under OGD when compared to pIRES controls.

However, we observed that in cells overexpressing renin-a, the increase in the rate of apoptosis marker-positive cells as well as in the rate of early apoptosis (caspase-positive, PI negative cells) under OGD was less prominent than in control cells. Under certain circumstances intra-cytoplasmic ANGII may decrease ROS production (see below), which would represent a protective effect. Indeed, the OGD-induced increase in ROS (percentage of MitoSOX positive cells) was slightly attenuated when compared with control cells. However, it remains obscure how ANGI is generated by renin-a or renin-b and how ANGIIP can be directed to the cytosol. Interestingly, under basal conditions enhanced levels of mitochondrial superoxides were observed not only in renin-a but also in renin-b overexpressing cells. A growing body of evidence indicates that the generation of ROS by mitochondria plays a critical role not only in the initiation of cell death during ischemia-reperfusion [22], but also in the activation of cell survival programs during pre- and postconditioning [23,24]. Although basal mitochondrial ROS production, which is associated with increased mitochondrial apoptosis, was increased in renin-b cells, the OGD-induced increase in the number of Annexin V-positive and caspase-positive apoptotic cells was markedly attenuated. In case of renin-b, extracellular generation of ANG II as the cause for ROS generation is unlikely since renin-b is not secreted. However, direct mitochondrial effects of ANG II have been demonstrated in various tissues [25]. ANG II applied to isolated mitochondria of neuronal cells increased mitochondrial ROS production via Nox4 and the AT1R, thereby increasing mitochondrial respiration [26]. On the other hand, ANG II was shown to decrease mitochondrial respiration via ROS and NO production mediated by mitochondrial AT2R [25,26]. In line with this, we observed an increased maximal respiration rate in renin-b overexpressing cells, which was accompanied by an increased spare capacity and thus enhanced stress tolerance [11]. Taken together, our data support the view of a protective cytosolic–mitochondrial RAS under starvation conditions caused by renin-b overexpression.

Shinohara et al. [27] argued that renin-b may inhibit renin-a expression thereby reducing effects of renin-a in the brain. However, in our hands the overexpression of renin-b did not decrease expression of renin-a in previous studies in H9c2 cells [10,16], but were protective, nevertheless. This argues for a specific role of renin-b in the protective process which is not mediated by downregulation of renin-a mRNA. Additionally, under glucose depletion, anoxia or OGD endogenous renin-b mRNA increased but renin-a mRNA did not decrease simultaneously [4,10]. This argues against a permanent invers regulation of the transcripts. Furthermore, some effects of renin-b appear to be independent of renin activity because they cannot be abolished by a renin enzyme inhibitor [16]. Thus, caspase-induced apoptosis was inhibited by the renin inhibitor CH732, whereas Annexin-induced apoptosis was not. Here, the mechanism of action is still unknown. A possible target of renin-b would be the cytosolic “renin binding protein” (RenBP) [28]. This protein exhibits N-acetyl-D-glucosamine 2-epimerase activity and therefore is also termed “NAGE”. Renin and NAGE form heterodimers particularly under depletion of high energy nucleotides such as ATP [29] thereby inhibiting each other’s activity. NAGE catalyzes the interconversion of N-acetyl-mannosamine to n-acetyl-D-glucosamine. The latter is needed for post-translational modifications of a variety of proteins, of which many are involved in metabolic processes, cell death and survival [30,31]. However, the hypothesis of an interaction of renin-b with RenBP/NAGE and its role for cellular survival still needs to be proven. Furthermore, renin (a or b) may bind the (pro)renin receptor (gene: *ATP6AP2*). The encoded protein is part of the canonical and non-canonical Wnt pathways and an associated subunit of v-ATPases. Thus, it exhibits a variety of roles in the cell cycle, differentiation processes and cellular homeostasis (for review see: [32]).

This is the first report on the overexpression of the complete renin-b transcript as it is present in vivo in the heart and upregulated under cardiac infarction exerts prominent protective effects under ischemia related conditions such as glucose depletion and anoxia. Thus, the hypothesis of a protective system based on renin-b expression is supported. Renin-b may act in a preconditioning-like manner. In addition, we exclude that the renin-a transcript, when overexpressed in H9c2 cells, exerts such protective effects, though it also contains the putative translation start site in exon 2. We are aware of the fact that H9c2 cells may not necessarily reflect the situation in differentiated cardiomyocytes of adult heart, but only the situation of immature precursor cells. Nevertheless, some of the beneficial effects of renin-b were also present in adult rat cardiomyocytes [9].

## Figures and Tables

**Figure 1 cells-10-01204-f001:**
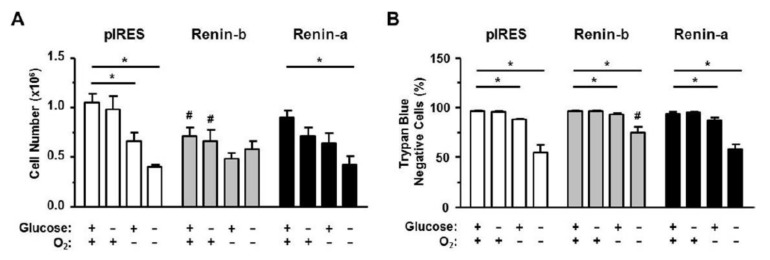
(**A**) Cell number and (**B**) percentage of trypan blue negative, viable cells of pIRES. control (n = 15), renin-b (n = 15) and renin-a cell lines (n = 11) overexpressing cytosolic and secretory renin, respectively. Cells were exposed to basal (non-treated) conditions, glucosestarvation alone, anoxia alone, and the combination of glucose starvation and anoxia (OGD) for 24 h. Renin-b overexpression was associated with a basal decrease in cell number without affecting the membrane integrity. Anoxia and OGD exposure resulted in a decrease 12 in cell number only in pIRES and renin-a cells and a reduction of membrane integrity in all three cell lines. However, for renin-b cells this effect was comparatively moderate. Data represent mean values ± SEM of 11–15 experiments. Differences were considered significant using two-way ANOVA with Bonferroni correction (*p*-values see text). (*): effects of the intervention (medium conditions) within the same cell line; (#): effects of renin-b or renin-a 17 overexpression, respectively, compared with pIRES control cells on same intervention.

**Figure 2 cells-10-01204-f002:**
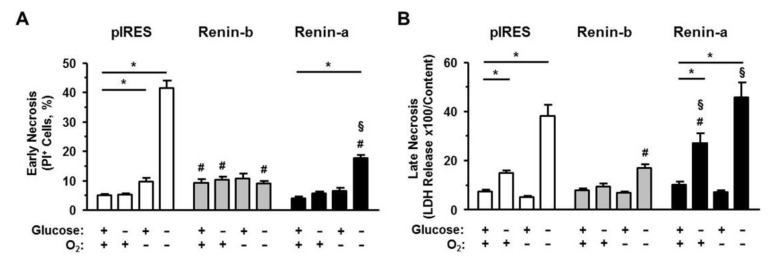
Renin-b protects H9c2 cells from necrosis induced by ischemia-related conditions. (**A**) Early and (**B**) late necrosis of pIRES control, renin-b and renin-a cell lines overexpressing cytosolic and secretory renin, respectively. Cells were exposed to basal (non-treated) conditions, glucose starvation alone, anoxia alone and the combination of glucose starvation and anoxia (OGD) for 24 h. Early necrosis was measured by PI labeling, where early necrotic cells were PI-positive but apoptosis-negative. Late necrosis was determined by the Cytotoxicity Detection Kit as declared. Renin-b overexpression was associated with an increased percentage of early necrotic cells under basal and glucose starvation conditions. OGD exposure resulted in a strong increase of early and late necrosis in pIRES control and renin-a cells, while in renin-b cells the amount of late necrosis increased only moderately. Data represent mean values ± SEM of 5–6 experiments. Differences were considered significant using two-way ANOVA with Bonferroni correction (*p*-values see text). (*) effects of the intervention (medium conditions) within the same cell line; (#): effects of renin-b or renin-a overexpression, respectively, compared with pIRES control cells on same intervention; (§): effects of renin-a overexpression compared with renin-b cells on same intervention.

**Figure 3 cells-10-01204-f003:**
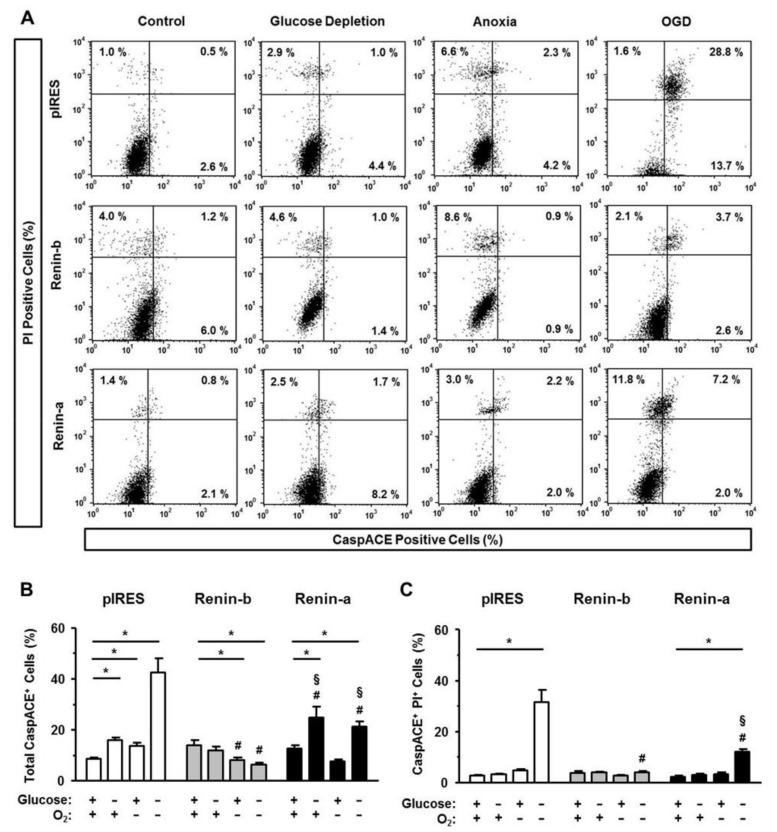
Renin-b protects H9c2 cells from activation of caspases induced by ischemia-related conditions. (**A**) Representative histograms of apoptotic cells exhibiting caspase activation (CaspACE marker). Propidium iodide (PI) labeling enabled differentiation between early apoptotic (PI-) and late apoptotic cells (PI+). (**B**) Mean percentages of total and (**C**) late apoptotic pIRES control, renin-b and renin-a cells are shown. Cells were exposed to basal (non-treated) conditions, glucose starvation alone, anoxia alone and the combination of glucose starvation and anoxia (OGD) for 24 h. Renin-b overexpression was associated with a protection against anoxia- and ODG-induced activation of caspases, which was observed in pIRES control and renin-a cells. Data represent mean values ± SEM of 6–7 experiments. Differences were considered significant using two-way ANOVA with Bonferroni correction (*p*-values see text). (*): effects of the intervention (medium conditions) within the same cell line; (#): effects of renin-b or renin-a overexpression, respectively, compared with pIRES control cells on same intervention; (§): effects of renin-a overexpression compared with renin-b cells on same intervention.

**Figure 4 cells-10-01204-f004:**
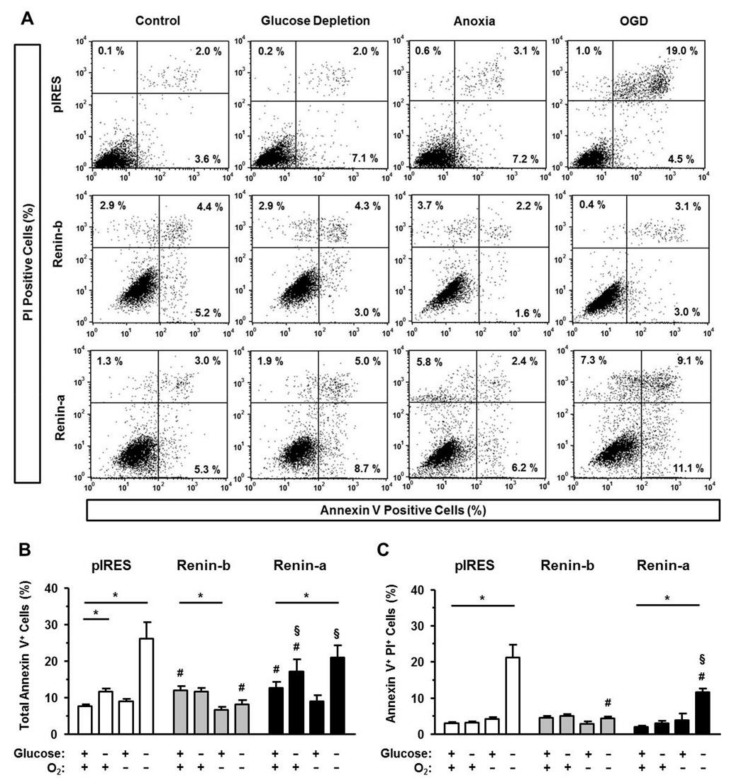
Renin-b protects H9c2 cells from intrinsically mediated apoptosis induced by ischemia-related conditions. (**A**) Representative histograms of apoptotic cells exhibiting Annexin V binding. Propidium iodide (PI) labeling enabled differentiation between early apoptotic (PI-) and late apoptotic cells (PI+). (**B**) Mean percentages of total and (**C**) late apoptotic pIRES control, renin-b and renin-a cells are shown. Cells were exposed to basal (non-treated) conditions, glucose starvation alone, anoxia alone, and the combination of glucose starvation and anoxia (OGD) for 24 h. Renin-b overexpression was associated with an increased number of total Annexin V-positive cells under basal conditions but also with a protection against glucose starvation- and ODG-induced apoptosis, which was observed in pIRES control and renin-a cells. Data represent mean values ± SEM of 6–7 experiments. Differences were considered significant using two-way ANOVA with Bonferroni correction (*p*-values see text). (*): effects of the intervention (medium conditions) within the same cell line; (#): effects of renin-b or renin-a overexpression, respectively, compared with pIRES control cells on same intervention; (§): effects of renin-a overexpression compared with renin-b cells on same intervention.

**Figure 5 cells-10-01204-f005:**
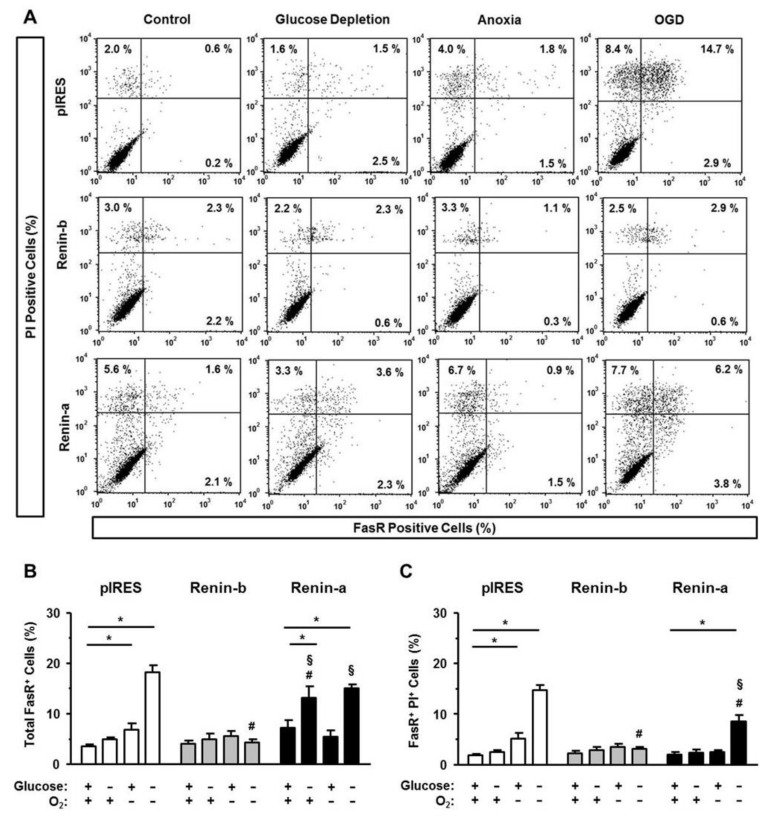
Renin-b protects H9c2 cells from Fas receptor (FasR)-mediated extrinsic pathway of apoptosis induced by ischemia-related conditions. (**A**) Representative histograms of apoptotic cells exhibiting FasR expression. Propidium iodide (PI) labeling enabled differentiation between early apoptotic (PI−) and late apoptotic cells (PI+). (**B**) Mean percentages of total and (**C**) late apoptotic pIRES control, renin-b and renin-a cells are shown. Cells were exposed to basal (non-treated) conditions, glucose starvation alone, anoxia alone and the combination of glucose starvation and anoxia (OGD) for 24 h. Renin-b overexpression was associated with a protection against anoxia- and ODG-induced apoptosis, which was observed in pIRES control and renin-a cells. Renin-a overexpression was accompanied by an increase in FasR expression during glucose starvation. Data represent mean values ± SEM of 6–7 experiments. Differences were considered significant using two-way ANOVA with Bonferroni correction (*p*-values see text). (*): effects of the intervention (medium conditions) within the same cell line; (#): effects of renin-b or renin-a overexpression, respectively, compared with pIRES control cells on same intervention; (§): effects of renin-a overexpression compared with renin-b cells on same intervention.

**Figure 6 cells-10-01204-f006:**
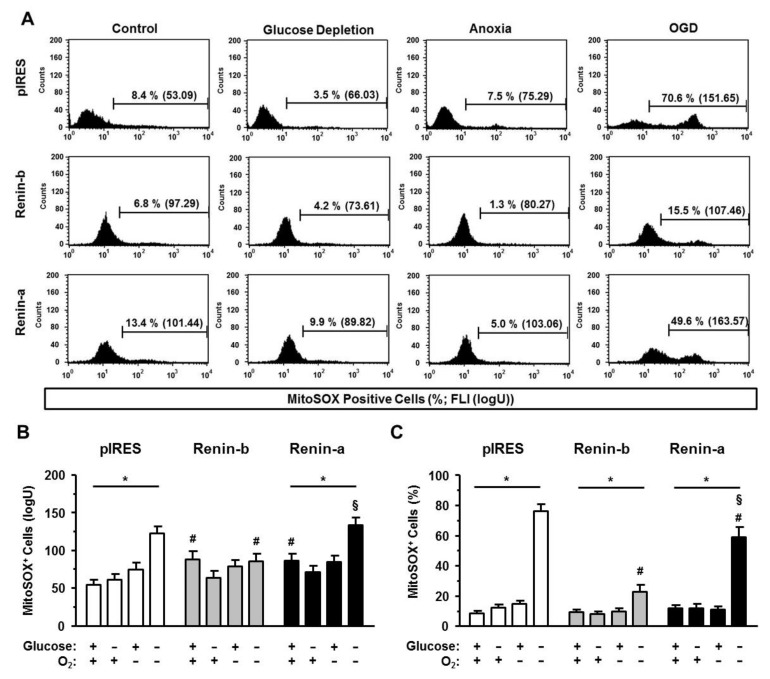
Renin-b protects H9c2 cells from accumulation of mitochondrial superoxides induced by OGD. (**A**) Representative histograms, (**B**) mean fluorescence intensities (FLI), and (**C**) mean percentages of MitoSOX-positive pIRES control, renin-b, and renin-a cells. Cells were exposed to basal (non-treated) conditions, glucose starvation alone, anoxia alone and the combination of glucose starvation and anoxia (OGD) for 24 h. Then, cells were incubated with the MitoSOX fluorophore to detect mitochondrially localized superoxides. Overexpression of renin-b and renin-a was associated with an increase in basal mitochondrial superoxides as detected by an increase in the mean FLI. Under OGD, the degradation of mitochondrial superoxides was inadequate leading to a marked increase in the percentage of MitoSOX-positive cells in pIRES control and renin-a cells. In renin-b cells, the number of MitoSOX-positive cells was also increased, but to a much smaller degree. Data represent mean values ± SEM of 9 experiments. Differences were considered significant using two-way ANOVA with Bonferroni correction (*p*-values see text). (*): effects of the intervention (medium conditions) within the same cell line; (#): effects of renin-b or renin-a overexpression, respectively, compared with pIRES control cells on same intervention; (§): effects of renin-a overexpression compared with renin-b cells on same intervention.

**Figure 7 cells-10-01204-f007:**
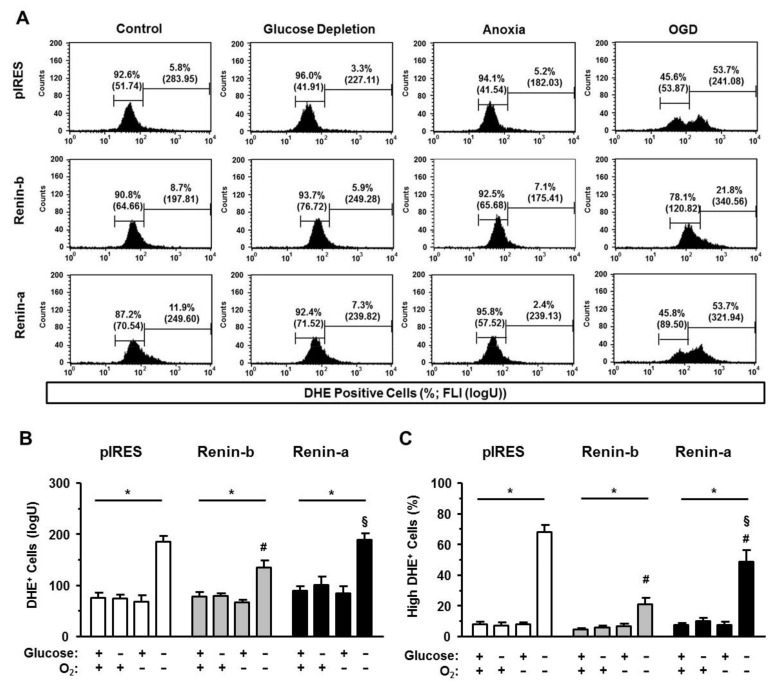
Renin-b protects H9c2 cells from accumulation of cytosolic reactive oxygen species (ROS) induced by OGD. (**A**) Representative histograms, (**B**) mean fluorescence intensities (FLI) and (**C**) mean percentages of dihydroethidium (DHE)-positive pIRES control, renin-b, and renin-a cells. Cells were exposed to basal (non-treated) conditions, glucose starvation alone, anoxia alone, and the combination of glucose starvation and anoxia (OGD) for 24 h. Then, cells were incubated with the DHE fluorophore to detect cytosolically localized ROS. Under OGD, the ROS content was increased as detected by an increased mean FLI in a subpopulation (DHEhigh-positive cells) of pIRES control cells. In renin-b and renin-a cells exposed to OGD the percentage of DHEhigh-positive cells was reduced compared to pIRES control cells. Data represent mean values ± SEM of 9 experiments. Differences were considered significant using two-way ANOVA with Bonferroni correction (*p*-values see text). (*): effects of the intervention (medium conditions) within the same cell line; (#): effects of renin-b or renin-a overexpression, respectively, compared with pIRES control cells on same intervention; (§): effects of renin-a overexpression compared with renin-b cells on same intervention.

**Figure 8 cells-10-01204-f008:**
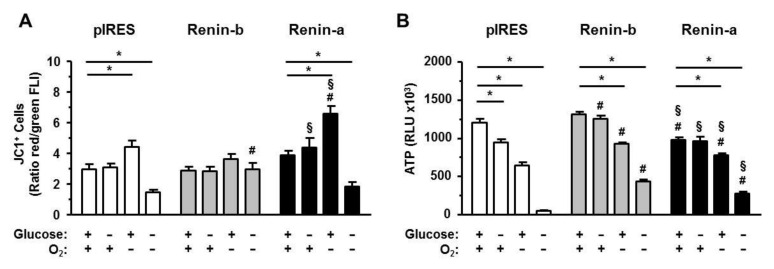
Renin-b protects H9c2 cells from OGD-induced collapse of mitochondrial membrane potential (ΔΨm) and mitigates anoxia- and OGD-induced ATP depletion. pIRES control, renin-b and renin-a cells were exposed to basal (non-treated) conditions, glucose starvation alone, anoxia alone and the combination of glucose starvation and anoxia (OGD) for 24 h. Afterwards, (**A**) ΔΨm was detected by incubating cells with the JC-1 dye (n = 6–10). Analysis of the ratio of red (disturbed ΔΨm) to green (intact ΔΨm) fluorescence intensity (FLI) of JC-1-positive cells represents the ΔΨm. OGD caused a marked decrease in ΔΨm in pIRES control and renin-a cells from which renin-b cells were completely spared. (**B**) ATP levels (n = 9) were detected by the CellTiter-Glo^®^ Luminescent Cell Viability Assay (n = 9). ATP levels were decreased in all cell lines during anoxia and OGD but mitigated in renin-b and renin-a cells. Data represent mean values ± SEM of indicated experiments. Differences were considered significant using two-way ANOVA with Bonferroni correction (*p*-values see text). (*): effects of the intervention (medium conditions) within the same cell line; (#): effects of renin-b or renin-a overexpression, respectively, compared with pIRES control cells on same intervention; (§): effects of renin-a overexpression compared with renin-b cells on same intervention.

**Table 1 cells-10-01204-t001:** Primer sequences for detection of transcript abundances.

Transcript	Forward Primer	Reverse Primer
renin exon (1–9)	exon 1: CCAGATGGGCGGGAGGAGGATG	exon 2: ATGAATTCACCCCATTCAGC
exon (1A-9)	exon 1A: TGAATTTCCCCAGTCAGTGAT	exon 2: GAATTCACCCCATTCAGCAC
Ywhaz	exon 2–3: CATCTGCAACGACGTACTGTCTCT	exon 3–4 GACTGGTCCACAATTCCTTTCTTG

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
