# Peer review of "Overexpression of Transcripts Coding for Renin-b but Not for Renin-a Reduce Oxidative Stress and Increase Cardiomyoblast Survival under Starvation Conditions"

_cells, 2021, doi:10.3390/cells10051204_

Round 1

Reviewer 1 Report

The study is well designed and interesting, includes new insights and I have a few suggestions and questions:

  1. By reviewing the manuscript through the system for determining plagiarism, the introduction and materials and methods should be written in their own words, there are a lot of prescribed sentence. Otherwise the rest is well written. Now, similarity index is 30%

  1. I like the introduction, it is simple and direct. But I think it should be stated in the introduction what is known so far about the connection between renin-a and renin-b and reactive oxygen species. In the introduction, you only state how you will measure it, but not what is the reason for that or what are the assumptions why it is important to measure it.

  1. Flow Cytometry - whether the cells were counted with trypan blue (under a microscope) or you also dyed the cells directly with viability dyes because cell viability dyes are critical controls for proper flow cytometry analysis? It often happens that some dead cells are stained and give a false positive result which can cause a wrong analysis of the results.

Author Response

The study is well designed and interesting, includes new insights and I have a few suggestions and questions:

  1. By reviewing the manuscript through the system for determining plagiarism, the introduction and materials and methods should be written in their own words, there are a lot of prescribed sentence. Otherwise the rest is well written. Now, similarity index is 30%

 Response: I agree that plagiarism is an important issue. The introduction and methods are written in our own words; however, some text passages can be found already in previous own publications. We now cite our previous publication and delete the text accordingly to reduce similarities.

With respect to the introduction, we do not feel it appropriate to change our own formulations that are optimized. We do not agree that this could be regarded as “plagiarism” (or “self-plagiarism”), since introductions do not contain new information, but just introduce the topic and summarize previous findings. Moreover, when working with the same topic after several similar introductions it becomes more and more difficult to find another way to write it.

  1. I like the introduction, it is simple and direct. But I think it should be stated in the introduction what is known so far about the connection between renin-a and renin-b and reactive oxygen species. In the introduction, you only state how you will measure it, but not what is the reason for that or what are the assumptions why it is important to measure it.

 Response: Thank you very much and thank you for your suggestions. We now provide the mechanism of how ROS leads to mitochondrial damage and cell death in the introduction. We cite the following papers: Webster (2012), Future Cardiology 8, 863-884; Hernandez-Resendiz (2018), Curr. Med. Chem. 25, 1275–1293, Green (1998), Sci. 281, 1309–1312 (1998), Miura (2012), Cardiovascular ResearchCardiovasc. Res. 94, 181–189.

  1. Flow Cytometry - whether the cells were counted with trypan blue (under a microscope) or you also dyed the cells directly with viability dyes because cell viability dyes are critical controls for proper flow cytometry analysis? It often happens that some dead cells are stained and give a false positive result which can cause a wrong analysis of the results.

Response: It is true that under some conditions, trypan blue-positive cells are not dead but exhibit only an impaired membrane integrity and are able to recover. Concerning trypan blue-positive cells we replace “dead cells” by “cells containing a disturbed membrane integrity”.

Reviewer 2 Report

The manuscript “Transcripts Coding for Non-Secretory Renin but Not for Secretory Renin Reduce Oxidative Stress and Increase Cardiomyoblast Survival under Starvation Conditions” by Wanka et al is a well-performed research article investigating the mechanism / role of non-secretory renin in cardiomyoblasts. 

In this paper, the authors overexpressed the full-length Renin-B and Renin-A transcripts in H9c2 cells and exposed these cells to various ischemia conditions.  They found that overexpression of Renin-B protected H9c2 cells against ischemia-related damage, whereas overexpression of Renin-A did not (and in some cases were actually harmful).  First, Renin-B overexpression protected cells against oxygen and glucose depletion (whereas overexpression of Renin-A did not).  Second, there was decreased early and late necrosis in Renin-B overexpression cells compared to control and Renin-A overexpression cells in the setting of oxygen and glucose depletions.  Third, overexpression of Renin-B was protective against apoptosis.  Fourth, overexpression of Renin-B prevented the accumulation of mitochondrial superoxides and cytosolic reactive oxygen species in the setting of oxygen / glucose depletion.  Finally, overexpression of Renin-B prevented disruption of the mitochondrial membrane potential and decreases in ATP levels following oxygen / glucose depletion.

While this paper is well-designed and well-executed, it is difficult to understand how this work advances on the author’s prior publications.  This paper would benefit from several changes and clarifications.      

Major Issues:

  1. I believe the authors need to more clearly state how this paper advances our understanding of Renin-B expression and the protective effects in cardiomyoblasts. It seems that many of the conclusions made in this work (effects on apoptosis, necrosis, ROS accumulation, mitochondrial membrane potential, ATP levels) were previously published by this group (Wanka et al.  Scientific Reports.  2020).  There are two main differences in this new work:  1-the full length Renin B transcript was overexpressed as opposed to just exons(2-9) and 2-the full length Renin A transcript was included as a control.  The authors need to clearly state how this current work provides an advance on what they have already published. 

  1. Along the same lines, it would be helpful if the authors address the significance of the protective effect of Renin B versus exon(2-9)renin. Does having the full length Renin B transcript provide more protection than just the exon(2-9)renin transcript?  Is the data from their two studies comparable?  Or at least, could this be considered in the discussion? 

  1. How much of the 5’UTR is included in this model? Is the traditional renin enhancer included in the 5’UTR?  It would be very interesting to know if the renin-A and renin-B transcripts share regulatory elements.

  1. The terminology for the alternative renin transcripts is confusing. Are “cytosolic renin”, non-secretory renin”, “exon(1A-9)renin” and “Renin B” all synonymous?  Is it known if the non-secretory renin described in this work is the same transcript that has been found in the rat brain and termed “Renin B”?

  1. Are these effects only for cardiomyoblasts, or would you expect to see the same findings in other cell types / cell lines? Have other cell lines been examined?

  1. Is the degree of Renin B overexpression physiologic? How do the levels of Renin B in this paper compare to the levels of Renin B seen in response to ischemia as shown in previous reports by this group (Clausmeyer S et al.    2000)?

  1. The authors speculate that the harmful effects of Renin-a are most likely explained by the production of angiotensin. However, could there be other possibilities?  What is the effect of alternative renin transcript overexpression on the native renin isoforms?  In other words, in the setting of renin-B overexpression, is the expression of renin-A decreased (and vice versa)?  Could this be a possible mechanism of the harmful effects of renin-A overexpression (feedback inhibition of renin-B and loss of renin-B’s protective effects)?

Minor Issues:

  1. What is “renin-c” (mentioned in the abstract but not defined or discussed further)?

  1. On line 32, is the alternative promotor located in “Intron A” (as written) or in “Intron 1”. It is my understanding that the renin-b transcript begins in Intron 1 which is now also known as “Exon 1A”. 

  1. Please make sure that scientific notation is used correctly (lines 86, 113, 126, 130, 142, etc).

  1. Line 101, I believe “death” should be “dead”.   

  1. There are misspellings on line 265 and 427.

Author Response

The manuscript “Transcripts Coding for Non-Secretory Renin but Not for Secretory Renin Reduce Oxidative Stress and Increase Cardiomyoblast Survival under Starvation Conditions” by Wanka et al is a well-performed research article investigating the mechanism / role of non-secretory renin in cardiomyoblasts. 

In this paper, the authors overexpressed the full-length Renin-B and Renin-A transcripts in H9c2 cells and exposed these cells to various ischemia conditions.  They found that overexpression of Renin-B protected H9c2 cells against ischemia-related damage, whereas overexpression of Renin-A did not (and in some cases were actually harmful).  First, Renin-B overexpression protected cells against oxygen and glucose depletion (whereas overexpression of Renin-A did not).  Second, there was decreased early and late necrosis in Renin-B overexpression cells compared to control and Renin-A overexpression cells in the setting of oxygen and glucose depletions.  Third, overexpression of Renin-B was protective against apoptosis.  Fourth, overexpression of Renin-B prevented the accumulation of mitochondrial superoxides and cytosolic reactive oxygen species in the setting of oxygen / glucose depletion.  Finally, overexpression of Renin-B prevented disruption of the mitochondrial membrane potential and decreases in ATP levels following oxygen / glucose depletion.

While this paper is well-designed and well-executed, it is difficult to understand how this work advances on the author’s prior publications.  This paper would benefit from several changes and clarifications.      

Major Issues:

  1. I believe the authors need to more clearly state how this paper advances our understanding of Renin-B expression and the protective effects in cardiomyoblasts. It seems that many of the conclusions made in this work (effects on apoptosis, necrosis, ROS accumulation, mitochondrial membrane potential, ATP levels) were previously published by this group (Wanka et al.  Scientific Reports.  2020).  There are two main differences in this new work:  1-the full length Renin B transcript was overexpressed as opposed to just exons(2-9) and 2-the full length Renin A transcript was included as a control.  The authors need to clearly state how this current work provides an advance on what they have already published. 
  1. Along the same lines, it would be helpful if the authors address the significance of the protective effect of Renin B versus exon(2-9)renin. Does having the full length Renin B transcript provide more protection than just the exon(2-9)renin transcript?  Is the data from their two studies comparable?  Or at least, could this be considered in the discussion? 

 Response to comments 1 and 2: Thank you very much for your suggestion. We have now pointed out more clearly our intention and new findings.

Introduction: To support the hypothesis of an existing endogenously protective renin-b, it was now essential to demonstrate that exactly this transcript, which is found in vivo, exerts the same protective effects as overexpression of exon(2-9), lacking exon1a and hence its 5’UTR. Therefore, our aims were (1) to investigate the effects of renin-a overexpression in H9c2 cells in general, (2) to demonstrate, that the already observed protective effects of artificial exon(2-9) renin overexpression were still present when the endogenous renin-b was used including its 5’UTR upstream of exons 2 on necrosis, apoptosis and the production of reactive oxygen species under starvation conditions such as glucose depletion and anoxia.

Discussion: We previously demonstrated protective effects of the overexpressed coding region of ren-in-b under glucose depletion and OGD. Renin-b attenuated and even prevented the in-crease in necrosis and apoptosis rates as well as ROS generation [8-11]. However, overexpressing only the coding region of renin-b is does not reflect the true endogenous situation, since the endogenous renin-b mRNA is comprising its 5’ untranslated region (5’UTR; about 80 bases of intron 1). This 5’UTR likely has regulatory functions and may even pre-vent the translation of a renin-b protein. To support the hypothesis of an existing endogenous system based on renin-b we had to show that the full-length renin-b transcript including its 5’UTR exerts the same protective effects. Here, we chose a degree of overexpression (5-fold), which is also found endogenously under stimulatory conditions, such as glucose depletion, anoxia, or OGD [4,10,11]. The protective effects of renin-b were still seen in the presence of its 5’UTR. Thus, the protective effects were independently of the 5’UTR. When indirectly comparing the effects of renin-b overexpression and exon(2-9) renin overexpression from our previous study, the effects were almost identical with respect to necrosis, apoptosis, ATP, ∆Ψm, and ROS levels. These data support our hypothesis of an existing endogenously protective system induced specifically by renin-b, but not by renin-a. We here need to consider, however, that the overexpression of renin-a was about twofold higher than the overexpression of renin-b. Thus, the effects are not directly comparable. Although we were not able to titrate down the overexpression of renin-a to exactly the level of overexpression of renin-b, we think it is unlikely that renin-a at lower dosage would be able to exert the same protective effects than renin-b, but cannot exclude it completely.

  1. How much of the 5’UTR is included in this model? Is the traditional renin enhancer included in the 5’UTR?  It would be very interesting to know if the renin-A and renin-B transcripts share regulatory elements.

 Response: For renin-b the complete 5’UTR is included (79 bases of intron 1). Its role, however, still remains to be investigated and also the role of the 5’UTR of renin-a.

Other than erroneously reported in this manuscript, the “renin-a” construct did not have its 5’UTR included. We apologize for this mistake. Nevertheless, our models still allows to obtain information about the coding part of renin-a. We agree that it would be of interest to specifically design a study to identify and compare the regulatory elements within the 5’UTRs of renin-a and renin-b.

Changes made: Materials and Methods: H9c2 cells were transfected as previously described [8] with a the empty pIRES vector either without or with exon(1A-9)renin or exon(1-9)renin or with the pIRES vector containing the coding region of renin-a or the coding region of renin-b together with its 5’UTR.

Results, 3.1. First sentence deleted.

  1. The terminology for the alternative renin transcripts is confusing. Are “cytosolic renin”, non-secretory renin”, “exon(1A-9)renin” and “Renin B” all synonymous?  Is it known if the non-secretory renin described in this work is the same transcript that has been found in the rat brain and termed “Renin B”?

 Response: yes, the terms are identical. Sorry for the confusion. We now use renin-a and renin-b unless we specially need to point out, which exons are present. Also, it is known that the transcript investigated in this work is the same than renin-b. Lee-Kirsch discovered this transcript in rats and mice (see ref. 2) at the same time than we discovered it in the rat (although we used another name for it, namely “exon1A-renin” see ref.1). However, we demonstrated that renin-b is not brain specific (see ref. 5). Changes made: We now use renin-b only, unless the exons needed to be specified.

  1. Are these effects only for cardiomyoblasts, or would you expect to see the same findings in other cell types / cell lines? Have other cell lines been examined?

 Response: Our guess is, that these effects are also present in other cell lines and we are presently investigating this hypothesis. However, we already know that renin-b overexpression (without 5’UTR) is protective under glucose depletion in primary rat cardiomyocytes (see last sentence of the discussion).

  1. Is the degree of Renin B overexpression physiologic? How do the levels of Renin B in this paper compare to the levels of Renin B seen in response to ischemia as shown in previous reports by this group (Clausmeyer S et al.    2000)?

 Response: The degree of overexpression very much resembles the endogenous renin mRNA seen under glucose depletion, hypoxia or OGD (endogenous renin-b mRNA increased four-fold by glucose depletion see Lutze et al., Cell Physiol Biochem (2017);42:1447-1457 and about tenfold by OGD. Wanka et al., Sci Reports (2020) 10:2329).

Changes made: Discussion: Here, we chose a degree of overexpression (5-fold), which is also found endogenously under stimulatory conditions, such as glucose depletion, anoxia, or OGD [4,10,11]. The protective effects of renin-b were still seen in the presence of its 5’UTR. Thus, the protective effects were independently of the 5’UTR. When indirectly comparing the effects of renin-b overexpression and exon(2-9) renin overexpression from our previous study, the effects were almost identical with respect to necrosis, apoptosis, ATP, ∆Ψm, and ROS levels. These data support our hypothesis of an existing endogenously protective system induced specifically by renin-b, but not by renin-a.

  1. The authors speculate that the harmful effects of Renin-a are most likely explained by the production of angiotensin. However, could there be other possibilities? 

Response: Thank you for your question. There is another possibility, namely the interaction of renin with the (pro)renin receptor. We have now discussed this possibility.

Changes: Discussion: . Furthermore, renin (a or b) may bind the (pro)renin receptor (gene: ATP6AP2). The en-coded protein is part of the canonical and non-canonical Wnt pathways and an associat-ed subunit of v-ATPases. Thus, it exhibits a variety of roles in the cell cycle, differentiation processes and cellular homeostasis (for review see: [32].

  1. What is the effect of alternative renin transcript overexpression on the native renin isoforms?  In other words, in the setting of renin-B overexpression, is the expression of renin-A decreased (and vice versa)?  Could this be a possible mechanism of the harmful effects of renin-A overexpression (feedback inhibition of renin-B and loss of renin-B’s protective effects)?

 Response: This is an important question. We are extending our discussion in this respect as follows:

Discussion: Shinohara et al [27] argued that renin-b may inhibit renin-a expression thereby reducing effects of renin-a in the brain. However, in our hands the overexpression of renin-b did not decrease expression of renin-a in previous studies in H9c2 cells [10,16], but were protective, nevertheless. This argues for a specific role of renin-b in the protective process which is not mediated by downregulation of renin-a mRNA. Also, under glucose depletion, anoxia or OGD endogenous renin-b mRNA increased but renin-a mRNA did not decrease simultaneously [4,10]. This argues against a permanent invers regulation of the transcripts. Furthermore,…

Minor Issues:

  1. What is “renin-c” (mentioned in the abstract but not defined or discussed further)?

 Response: renin-c is an apparently lung specific isoform, which also has its first in frame AUG in exon 2.

Changes: Introduction: Alternative renin transcripts, lacking exon 1, have been identified in several tissues including the heart of rats (termed exon1a renin) [1], in the brain of mice (termed renin-b) [2] as well as in the brain (renin-b) or lung (renin-c) of transgenic mice expressing a human renin gene construct [3].

  1. On line 32, is the alternative promotor located in “Intron A” (as written) or in “Intron 1”. It is my understanding that the renin-b transcript begins in Intron 1 which is now also known as “Exon 1A”. 

 Response: The alternative promoter is located in intron 1. Intron 1 and intron A were used synonymously. It may be better indeed to use intron A only for the alternative intron 1 of renin-b (between the 5’UTR = previously termed exon1A and exon 2). We have done so now.

  1. Please make sure that scientific notation is used correctly (lines 86, 113, 126, 130, 142, etc).

  Response: the superscriptions were lost during editing. We have now restored them.

  1. Line 101, I believe “death” should be “dead”.   

 Response: thank you, we have replaced death by dead where necessary.

Reviewer 3 Report

In the manuscript, the authors studied the cytoprotective effect of non-secretary renin (renin-b) against oxygen-glucose deprivation in vitro. Overall, the experiments are well performed and the results are clear.

However, I have several points, which I think that the authors should address before acceptance of the paper for publication. Specific comments are below.

Major points.

  1. The authors show only the expression levels in the transient expression system, which are described in materials and methods; renin-a and renin-b levels increased 4-hold and 10-fold, respectively. Throughout the study, the authors used stable lines that express renin-a and renin-b, respectively. However, the expression levels of each clone are not shown in the text. This should be presented in the result section.
  2. Regarding above, since these expression levels directly contribute to the protective effects, the authors should compare clones which have “almost similar levels” of renin-a and renin-b.
  3. As I see the results, renin-a also exhibits a cytoprotection against apoptosis or early necrosis against oxygen-glucose deprivation. However, there is no discussion about the protective effect of renin-a. How does renin-a work although it is secreted to extracellular space? Clarification is needed in the discussion section.
  4. Line 183–184; More direct evidences are needed to state that renin-b cells exhibit a growth arrest during exposure to ischemia-related conditions. For example, the authors should analyze DNA contents with flow cytometry or perform an EdU/BrdU staining.

Minor points.

  1. Abstract (line 11): renin-c is not explained in the main text.
  2. Table 1: I cannot find any data using primers for exon (2-9) or Ywhaz.
  3. English should be carefully revised.

Author Response

In the manuscript, the authors studied the cytoprotective effect of non-secretary renin (renin-b) against oxygen-glucose deprivation in vitro. Overall, the experiments are well performed and the results are clear.

However, I have several points, which I think that the authors should address before acceptance of the paper for publication. Specific comments are below.

Major points.

  1. The authors show only the expression levels in the transient expression system, which are described in materials and methods; renin-a and renin-b levels increased 4-hold and 10-fold, respectively. Throughout the study, the authors used stable lines that express renin-a and renin-b, respectively. However, the expression levels of each clone are not shown in the text. This should be presented in the result section.

Response: We are still using the same lines, not stable ones. The term “stable”, was used incorrectly and did not meet our intention.

Changes made: Methods: we deleted “stable”.

  1. Regarding above, since these expression levels directly contribute to the protective effects, the authors should compare clones which have “almost similar levels” of renin-a and renin-b.

Response: We agree. However, it was impossible to reach such a goal. Overexpression of renin-a was higher than that of renin-b. We are now considering this fact in the discussion:

Changes made: Discussion: We here need to consider, however, that the overexpression of renin-a was about twofold higher than the overexpression of renin-b. Thus, the effects are not directly comparable. Although we were not able to titrate down the overexpression of renin-a to exactly the level of overexpression of renin-b, we think it is unlikely that renin-a at lower dosage would be able to exert the same protective effects than renin-b, but cannot exclude it completely.

  1. As I see the results, renin-a also exhibits a cytoprotection against apoptosis or early necrosis against oxygen-glucose deprivation. However, there is no discussion about the protective effect of renin-a. How does renin-a work although it is secreted to extracellular space? Clarification is needed in the discussion section.

Response: Thank you for your comment. This is an interesting observation, that we did not consider with appropriate respect. We have now pointed out more clearly a putative protective effect of renin-a and discussed it. The lower rates of apoptosis argues for some protection by renin-a.

Changes made: Discussion: However, we observed that in cells overexpressing renin-a, the increase in the rate of apoptosis marker-positive cells as well as in the rate of early apoptosis (caspase-positive, PI negative cells) under OGD was less prominent than in control cells. Under certain cir-cumstances intra-cytoplasmic ANGII may decrease ROS production (see below), which would represent a protective effect. Indeed, the OGD-induced increase in ROS (percentage of MitoSOX positive cells) was slightly attenuated when compared with control cells However, it remains obscure how ANGI is generated by renin-a or renin-b and how AN-GII can be directed to the cytosol..

  1. Line 183–184; More direct evidences are needed to state that renin-b cells exhibit a growth arrest during exposure to ischemia-related conditions. For example, the authors should analyze DNA contents with flow cytometry or perform an EdU/BrdU staining.

Response: we agree and now do not speak about growth arrest any more.

Minor points.

  1. Abstract (line 11): renin-c is not explained in the main text.

Response: renin-c is an apparently lung specific isoform, which also has its first in frame AUG in exon 2.

Changes made: Introduction, first sentence.

  1. Table 1: I cannot find any data using primers for exon (2-9) or Ywhaz.

We have now included deleted this information from the table.

  1. English should be carefully revised.

Response: We apologize. Unfortunately, we were not able to send the manuscript to a professional English editor in time. Therefore, I would like to use the editing service of MDPI checking grammar, spelling, punctuation and style.

Round 2

Reviewer 3 Report

The authors have addressed all my concerns and therefore I have no further points.